# Genetic Evidence for a Mixed Composition of the Genus *Myoxocephalus* (Cottoidei: Cottidae) Necessitates Generic Realignment

**DOI:** 10.3390/genes11091071

**Published:** 2020-09-11

**Authors:** Evgeniy S. Balakirev, Alexandra Yu. Kravchenko, Alexander A. Semenchenko

**Affiliations:** 1A.V. Zhirmunsky National Scientific Center of Marine Biology, Far Eastern Branch, Russian Academy of Sciences, Vladivostok 690041, Russia; sasha_postbox@mail.ru; 2School of Biomedicine, Far Eastern Federal University, Vladivostok 690950, Russia; 3Laboratory of Ecology and Evolutionary Biology of Aquatic Organisms, School of Natural Sciences, Far Eastern Federal University, Vladivostok 690950, Russia; semenchenko_alexander@mail.ru

**Keywords:** mitochondrial genome, ribosomal DNA, *COI*, genetic divergence, fourhorn sculpin *Myoxocephalus quadricornis*, belligerent sculpin *Megalocottus platycephalus*, *Myoxocephalus*, *Megalocottus*, *Microcottus*, *Argyrocottus*, *Porocottus*, taxonomic realignment, Cottidae

## Abstract

Sculpin fishes belonging to the family Cottidae represent a large and complex group, inhabiting a wide range of freshwater, brackish-water, and marine environments. Numerous studies based on analysis of their morphology and genetic makeup frequently provided controversial results. In the present work, we sequenced complete mitochondrial (mt) genomes and fragments of nuclear ribosomal DNA (rDNA) of the fourhorn sculpin *Myoxocephalus quadricornis* and some related cottids to increase the power of phylogenetic and taxonomic analyses of this complex fish group. A comparison of the *My. quadricornis* mt genomes obtained by us with other complete mt genomes available in GenBank has revealed a surprisingly low divergence (3.06 ± 0.12%) with *Megalocottus platycephalus* and, at the same time, a significantly higher divergence (7.89 ± 0.16%) with the species of the genus *Myoxocephalus*. Correspondingly, phylogenetic analyses have shown that *My. quadricornis* is clustered with *Me. platycephalus* but not with the *Myoxocephalus* species. Completely consistent patterns of divergence and tree topologies have been obtained based on nuclear rDNA. Thus, the multi-gene data in the present work indicates obvious contradictions in the relationships between the *Myoxocephalus* and *Megalocottus* species studied. An extensive phylogenetic analysis has provided evidence for a closer affinity of *My. quadricornis* with the species of the genus *Megalocottus* than with the species of the genus *Myoxocephalus*. A recombination analysis, along with the additional GenBank data, excludes introgression and/or incorrect taxonomic identification as the possible causative factors responsible for the observed closer affinity between the two species from different genera. The above facts necessitate realignment of the genera *Myoxocephalus* and *Megalocottus*. The genetic data supports the two recognized genera, *Myoxocephalus* and *Megalocottus*, but suggests changing their compositions through transferring *My. quadricornis* to the genus *Megalocottus*. The results of the present study resolve the relationships within a complex group of sculpin fishes and show a promising approach to phylogenetic systematics (as a key organizing principle in biodiversity research) for a better understanding of the taxonomy and evolution of fishes and for supplying relevant information to address various fish biodiversity conservation and management issues.

## 1. Introduction

The genus *Myoxocephalus* Tilesius 1811 comprises 15 or 16 species [1,2,3,4] distributed widely from the northern Pacific and Arctic Oceans to the northern Atlantic Ocean [5,6,7,8,9]. Members of this genus have diverse morphologies but are clearly differentiated from other taxa in the family Cottidae [5,6,10,11,12,13]. Morpho-anatomical [5,6,14,15] and genetic [16,17,18,19] studies have shown that the genus *Myoxocephalus* is closely related to the genus *Megalocottus* Gill 1861 represented by the only species, the belligerent sculpin *Megalocottus platycephalus* (Pallas 1814). Mitochondrial DNA (mtDNA) markers [20] have revealed a genetic heterogeneity associated with the spatial distribution of the species which might result in the emergence of two subspecies known for *Me. platycephalus*: *Me. p. platycephalus* (Pallas 1814) and *Me. p. taeniopterus* (Kner 1868). Using osteological and myological characters, Yabe [13] found that the *Myoxocephalus*, *Microcottus* Schmidt 1940, *Argyrocottus* Herzenstein 1892, and *Porocottus* Gill 1859 constitute a group of closely related genera, which are referred to as the *Myoxocephalus* group. Further genetic studies confirmed the Yabe’s [13] classification [19,21,22], except *Argyrocottus*, which still remains unstudied by genetic methods.

The fourhorn sculpin *Myoxocephalus quadricornis* (Linnaeus 1758) is an arctic euryhaline species with circumpolar distribution, inhabiting shallow Arctic waters south to the Gulf of Anadyr, waters off St. Lawrence Island, northern Bristol Bay, and the Bering Sea in the Pacific, and from off northern Greenland to the Baltic Sea in the Atlantic [5,6,7,8,9]. The species is represented by several ecological forms, which complicates its taxonomic classification [23]. The taxonomic nomenclature of fourhorn sculpin based on morphological characters has been subject to various changes with multiple known synonyms. Formerly, this fish was referred to as *Triglopsis quadricornis* (Linnaeus 1758), *Cottus hexacornis* Richardson 1823, *Oncocottus hexacornis* (Richardson 1823), etc. [2,3,4].

Little is known about the population genetics and phylogenetics of *My. quadricornis*; available data are limited mostly to allozymes [24] and mitochondrial markers [25,26,27,28]. Using 30 enzyme loci, Gyllensten and Ryman [24] detected a low genetic heterogeneity between the Baltic freshwater and brackish-water populations with no evidence for deviations from the Hardy–Weinberg expectation, or linkage disequilibrium. In addition, the authors detected a statistically significant spatial and temporal allele frequency heterogeneity between the Baltic populations, with the most pronounced temporal allele frequency shifts found at highly polluted localities [24].

Using the *cytb*, *atp6*, and *atp8* mt genes, Kontula and Väinölä [25] investigated the spatial structure of *My. quadricornis* from the Arctic coastal waters and from ‘glacial relict’ populations in Nearctic and Palearctic postglacial lakes. The authors showed a principal phylogeographical split that separates the North American continental deepwater sculpin *My. q. thompsonii* from a lineage of the Arctic marine and North European landlocked populations of the fourhorn sculpin *My. quadricornis*. However, the Baltic Sea populations and the Fennoscandian lacustrine populations of *My. quadricornis* exhibit a significant genetic similarity [25]. Kontula et al. [26] used the *cytb*, *atp6*, and *atp8* mt sequences of *Triglopsis quadricornis* (a synonym of *My. quadricornis*, see above) and *Malacocottus zonurus* as outgroups to investigate the endemic diversification of the cottoid fishes from Lake Baikal. Hubert et al. [27] and Mecklenburg et al. [28] assessed the degree to which the *COI* gene-based DNA barcoding discriminates marine and freshwater fishes, including *My. quadricornis*. The authors concluded that fish species can be efficiently identified through DNA barcoding, and that the *COI* library can subsequently find application in ecology and systematics. McCusker et al. [29] noted, however, that sculpins (along with eelpouts and rocklings) proved to be among the most challenging groups for DNA barcoding, which could be explained by a number of factors including incomplete lineage sorting, introgressive hybridization, or erroneous taxonomic identifications.

Previously, we investigated complete mt genomes and taxonomic relationships of sculpin fishes including *Cottus szanaga*, *C. volki*, *C. amblystomopsis*, *C. czerskii*, *My. polyacanthocephalus*, *My. jaok*, and *Me. platycephalus* [30,31,32,33,34,35,36]. In the present study, the molecular phylogenetic analyses have been carried out using mitochondrial and nuclear DNA datasets and based on extensive taxon sampling, including most of the constituent species in the genera *Myoxocephalus* and *Megalocottus* (*My. quadricornis*, *My. ochotensis*, *My. scorpius*, *My. scorpioides*, *My. octodecemspinosus*, *My. aenaeus*, *My. q. thompsonii*, *My. polyacanthocephalus*, *My. brandtii*, *My. stelleri*, *My. jaok*, and *Me. platycephalus*) and related sculpin fishes (*Microcottus sellaris*, *Argyrocottus zanderi*, *Porocottus allisi*, *P. minutus*, and *P. japonicus*), with two main objectives: (1) investigate the phylogenetic affinities of different species of *Myoxocephalus* and (2) clarify the *Myoxocephalus*–*Megalocottus* taxonomic interrelationships within the family Cottidae, which would allow assessment of circumscription for the genera.

## 2. Materials and Methods

### 2.1. Fish Specimens and Sequencing

The environmental status of the fourhorn sculpin *My. quadricornis* is considered as stable; it is listed under Least Concern on the International Union for Conservation of Nature’s (IUCN) Red List of Threatened Species [37]. The other sculpin fishes studied here are not included in the IUCN Red List of Threatened Species; they are not listed as endangered, vulnerable, rare, or protected species in the Russian Federation, either. The sampling points are located beyond any protected areas. The sites of the field studies are not privately-owned or protected.

The specimens of *My. quadricornis* were collected from Sukhoe More Bay, Dvina Bay, White Sea (March 18, 2017; 64°55′53″ N, 40°17′19″ E). The specimens of *Argyrocottus zanderi* Herzenstein 1892, *Me. platycephalus*, *Myoxocephalus polyacanthocephalus* (Pallas 1814), *Myoxocephalus brandtii* (Steindachner 1867), *Myoxocephalus stelleri* Tilesius 1811, *Myoxocephalus jaok* (Cuvier 1829), *Microcottus sellaris* (Gilbert 1896), *Porocottus minutus* (Pallas 1814), *Porocottus allisi* (Jordan and Starks 1904), and *Porocottus japonicus* Schmidt 1935 were collected from the Sea of Japan and the Sea of Okhotsk (for localities and coordinates, see Appendix A). The genomic DNA was extracted using a KingFisher Flex System and a set of reagents of a MagMAX DNA Multi-Sample Kit (ThermoFisher Scientific, Waltham, MA, USA).

The complete mt genome and rDNA sequences were amplified in five and three overlapping fragments, respectively, using the Phusion High-Fidelity DNA Polymerase (ThermoFisher Scientific). Libraries were prepared using an Ion Plus Fragment Library Kit and unique adapters (Ion Xpress, Waltham, MA, USA) with pre-fragmentation on a Covaris M220 Focused ultrasonicator. Ready libraries were sequenced on an Ion S5 sequencing platform (ThermoFisher Scientific) at the Far Eastern Federal University (Vladivostok, Russia). The complete mt genomes obtained were initially annotated using the MitoFish Web Server [38] and further manually adjusted in MEGA 7 [39] by comparing them with mt genomes of other sculpin fishes. The rDNA sequences were aligned using SPAdes 3.14.1 [40] with a correction of IonTorrent data using the IonHammer tool available in the SPAdes software and annotated using sequences of other Scorpaeniformes fishes available in GenBank. The nucleotide sequences were deposited in GenBank under the accession numbers MT303953–MT303954, MT890585–MT890586, MT906795–MT906796, MT909822–MT909823 (mt genomes) and MT497857–MT497873 (rDNA). Additional complete mt genomes and separate *COI* gene sequences of sculpin fishes were accessed from the GenBank National Center for Biotechnology Information (NCBI) genetic sequence database [41] (see Appendix A for accession numbers).

### 2.2. DNA Sequence Analysis

The nucleotide sequences were aligned using the MUSCLE [42] and MAFFT v. 7 [43] software. The alignments were analyzed for evidence of recombination using various recombination detection methods provided in the RDP4 program [44]. The DnaSP v. 5 [45], PROSEQ v. 2.9 [46], and MEGA 7 [39] programs were used for interspecific analysis of divergence. Phylogeny reconstructions were based on the mt genome, rDNA, and *COI* gene alignments using the maximum-likelihood methods available in IQ-TREE [47,48]. ModelFinder [49] or jModelTest [50] were used to find the best-fit model of nucleotide substitution under the maximum likelihood criterion. The General Time Reversible + gamma + invariant sites (GTR + G +I; [51]), Tamura-Nei (TN93; [52]), and Hasegawa-Kishino-Yano (HKY; [53]) models showed the lowest Akaike Information Criterion (AIC; [54]) value (189,144.4265, 22,798.8538, and 6315.5052) and Bayesian information criterion (BIC; [55]) scores (189,529.5437, 23,075.7355, and 6625.4694) for the mt genome, rDNA, and *COI* gene alignments, respectively, and they were selected for further phylogenetic reconstructions. The ultrafast maximum likelihood bootstrap analysis [56] consisted of 1000 replicates. Partitioned analyses were performed in PartitionFinder 2.1.1 [57] that allowed the overall rate to vary between partitions including each protein-coding gene (see Appendix A for the best model of substitution under the maximum likelihood criterion).

The complete dataset was also analyzed by the Bayesian inference using MrBayes v. 3.2.7a (released March 6, 2019; [58]) under the GTR + G + I model. Analyses were performed as two independent runs, each with four incrementally heated Metropolis-coupled Monte-Carlo Markov Chains. Output trees and data were sampled for every 500 generations. Likelihood values reached a plateau within 15,000–25,000 generations. A total of 4002 trees in two files were read and 3002 of them were sampled. For the mt genomes, the log likelihood values increased from below −148,989.341 to around −94,582.436 in the first 5000 generations and then to around −94,479.814 after 1,000,000 generations. The likelihood of the best state for “cold” chain of run 1 was −94,452.62, and the likelihood of the best state for “cold” chain of run 2 was −94,452.71. The average standard deviation of split frequencies was 0.000824 after 1,000,000 generations, indicating stationary conditions. In convergence diagnostic, the Potential Scale Reduction Factor (PSRF) [59] was between 1.000 and 1.011 for all parameters, thus, indicating a good sample from the posterior probability distribution. The log likelihood values and standard deviations of split frequencies, as well as the values of convergence diagnostic for the rDNA and *COI* genes, are provided in the Text S1 file.

The tree topologies for the sculpin fishes obtained by the maximum-likelihood method and by the Bayesian inference were very similar. The close congruence can be explained by the fact that the three separate datasets were relatively straightforward and included the complete mt genomes, rDNA, and *COI* sequences of sculpin fishes only. The quality of alignment was high, and the length of alignment was long enough (a total of 19.1 kb for the mt genomes, 6.3 kb for the rDNA, and 1.1 kb for the *COI* gene). As has been shown by many authors, the relative efficiencies of different methods applied for obtaining correct tree topology are very close under these conditions [60]. The difference between the methods applied to the sculpin fishes (our data) resulted in the slightly different topologies and bootstrap values and did not cause changes in the relationships between *My. quadricornis* and its close relatives. To be conservative, we here provide the lowest bootstrap values obtained by the maximum-likelihood method.

## 3. Results

### 3.1. Complete Mitochondrial Genomes

The mt genomes (120× coverage) of *My. quadricornis*, *P. allisi*, and *A. zanderi* are 16,682 bp, 16,367–16,370, and 16,608, respectively. For *Microcottus sellaris*, only the main fragment (12,442) of the mt genome has been sequenced. The gene arrangement, composition, and size are very similar to those of the sculpin fish genomes published previously. A comparison of the mt genomes obtained by us with other complete mt genomes of several sculpin genera available in GenBank (*Myoxocephalus* Tilesius 1811, *Megalocottus* Gill 1861, *Enophrys* Swainson 1839, *Gymnocanthus* Swainson 1839, *Icelus* Krøyer 1845, *Cottiusculus* Schmidt in Jordan and Starks 1904, *Hemilepidotus* Cuvier 1829, *Comephorus* Lacepède 1800, *Mesocottus* Gratzianov 1907, *Trachidermus* Heckel 1837, and *Clinocottus* Gill 1861) has revealed a surprisingly close affinity (*D*_xy_ = 0.0306 ± 0.0012) of *My. quadricornis* to a non-congeneric sculpin species, *Me. platycephalus* (Figure 1; Cluster 1). The level of divergence between *My. quadricornis* and *Me. platycephalus* (3.06 ± 0.12%) matches the average value of divergence between different species within the same genus (intrageneric level) of cottids [16,17,18].

Another group of sequences (Figure 1; Cluster 2) contains the *Myoxocephalus* species only, including *My. polyacanthocephalus*, *My. jaok*, and *My. scorpius*. The level of divergence (*D*_xy_ = 0.0314 ± 0.0012) between the sequences within this group is slightly higher, but still matches the value of divergence for different species within the same genus (intrageneric level). The level of divergence between clusters 1 and 2 is significantly higher (*D*_xy_ = 0.0789 ± 0.0016), matching the values of divergence, albeit the lowest ones, between different genera within the family Cottidae (intrafamily level) [16,17,18]. Comparable values of divergence based on complete mt genomes have been obtained in the pairwise comparisons between the other genera of the cottids studied (Figure 1) such as *Megalocottus*–*Microcottus* (*D*_xy_ = 0.0829 ± 0.0024), *Myoxocephalus*–*Microcottus* (*D*_xy_ = 0.0865 ± 0.0015), *Myoxocephalus*–*Enophrys* (*D*_xy_ = 0.0898 ± 0.0025), *Myoxocephalus*–*Argyrocottus* (*D*_xy_ = 0.0910 ± 0.0020), *Megalocottus*–*Enophrys* (*D*_xy_ = 0.0970 ± 0.0023), *Megalocottus*–*Argyrocottus* (*D*_xy_ = 0.0987 ± 0.0019), *Myoxocephalus*–*Icelus* (*D*_xy_ = 0.1025 ± 0.0023), and *Myoxocephalus*–*Porocottus* (*D*_xy_ = 0.1032 ± 0.0021).

The species belonging to the genera *Argyrocottus* and *Porocottus* are clustered with the genus *Myoxocephalus* (Figure 1). The level of divergence (*D*_xy_) between *A. zanderi* and *P. allisi* (0.1093 ± 0.0021), as well as between these two species and the *Myoxocephalus* cluster 2 (0.0950 ± 0.0017), matches the value of divergence between different genera within the family Cottidae (intrafamily level). The species belonging to the genus *Microcottus* is clustered with the genus *Megalocottus* (Figure 1). The level of divergence between *M. sellaris* and the *Megalocottus* cluster 1 (0.0829 ± 0.0025) also matches the intrafamily level.

### 3.2. Nuclear Ribosomal Genes

The 6.3 kb fragment of the nuclear rDNA sequences of the genera *Myoxocephalus* and *Megalocottus* shows completely consistent patterns of divergence and tree topologies with those observed for the mt genomes. There are two clusters containing *Myoxocephalus* species (Figure 2; Cluster 1 and Cluster 2). The first cluster contains both *My. quadricornis* and *Me. platycephalus* with the difference between them being very low (*D*_xy_ = 0.0018 ± 0.0005). The second cluster contains the *Myoxocephalus* species only, including *My. jaok*, *My. stelleri*, *My. brandtii*, and *My. polyacanthocephalus*, with the average distance (*D*_xy_) equal to 0.0037 ± 0.0005. The difference between Clusters 1 and 2 is significantly higher (*D*_xy_ = 0.0071 ± 0.0010). The members of the genera *Microcottus* and *Porocottus* form separate clusters outside of both *Myoxocephalus* and *Megalocottus*, with relatively high genetic distances (*D*_xy_) varying from 0.0122 ± 0.0015 to 0.0160 ± 0.0017 and a significant bootstrap support (Figure 2). The tree topologies obtained for the *Myoxocephalus* + *Megalocottus* and *Microcottus* + *Porocottus* clusters are slightly different when based on the rDNA fragments (Figure 2) and mtDNA (Figure 1), which can reflect specific evolutionary trends in nuclear and mitochondrial genomes. The phylogenetic discordance between nuclear and mitochondrial genomes is not a rare phenomenon among fishes [61] and many other organisms [62].

Nuclear ribosomal genes are frequently used in molecular systematics [63,64]. Both small (18S) and large (28S) ribosomal genes, characterized by slow rate of evolution, have been used for phylogenetic studies on distantly related species to resolve macroevolutionary issues such as the origin of tetrapods and general vertebrate phylogeny [65]. Despite rDNA genes are less sensitive than mtDNA genes in analysis of close groups (species or genus level), nevertheless, data obtained using nuclear ribosomal markers are important for comparative purposes. As an independent source of genetic information, our rDNA data confirms the results obtained based on the mtDNA sequences.

### 3.3. GenBank COI Gene Dataset

As in the case of mt genome and rDNA data, the *COI* gene cluster 1 contains both the *Myoxocephalus* species (*My. scorpioides*, *My. octodecemspinosus*, *My. aenaeus*, and *My. quadricornis*) and *Me. platycephalus* (represented by two subspecies, *Me. p. taeniopterus* and *Me. p. platycephalus*; see [20]) (Figure 3). The average difference (*D*_xy_) between the species of Cluster 1 is 0.0387 ± 0.0047 (using three randomly picked sequences per species). The *COI* gene Cluster 2 contains the species belonging to the genus *Myoxocephalus* only, including *My. stelleri*, *My. ochotensis*, *My. jaok*, *My. polyacanthocephalus*, *My. brandtii*, and *My. scorpius* (Figure 3). The average difference (*D*_xy_) between the species of Cluster 2 is 0.0265 ± 0.0037. The average levels of divergence within both clusters are relatively low, varying within 2.7–3.9%, which fits into the range of interspecific divergence for sculpin fishes [16,17,18,36]. The level of divergence between Clusters 1 and 2 is significantly higher, *D*_xy_ = 0.0826 ± 0.0091, indicating the intergeneric level of divergence for cottids [16,17,18] (see also the Section 3.1). Thus, the *COI* dataset containing multiple GenBank sequences shows a significantly higher divergence (8.3%) between the clusters than within them (2.7–3.9%), which agrees with the analyses based on the complete mt genomes and the rDNA sequences.

The GenBank *COI* sequences demonstrate also a close similarity of the three other *Myoxocephalus* species, *My. aenaeus*, *My. octodecemspinosus*, and *My. scorpioides*, to the *My. quadricornis* plus *Me. platycephalus* cluster (Figure 3, Cluster 1). This suggests that the above-listed three species may also belong to the genus *Megalocottus*. The close affinity between *My. cf. scorpiodes* and *Me. platycephalus* was also reported previously by Knope [19] based on mitochondrial (*cytb*) and nuclear (the first intron of the nuclear S7 ribosomal protein) markers. However, the available data are still insufficient to draw any reliable conclusion.

### 3.4. Alternative Hypotheses: Recombination and Species Misidentification

The phylogenetic inconsistency that we detected may reflect the historical hybridization event(s) between the *Myoxocephalus* and *Megalocottus* species which could result in interspecific recombination of their mtDNA (as it has been found in other organisms including fishes; see, e.g., [66,67]), or it may be due to incorrect taxonomic identification. We, therefore, analyzed the mt genome alignments for evidence of recombination using various recombination detection methods provided in the RDP4 program [44]. All the methods failed to reveal any signal of recombination between the mt genomes of the *Myoxocephalus* and *Megalocottus* species, thus rejecting hybridization as a possible explanation for the anomalous similarity between the *My. quadricornis* and *Me. platycephalus* mt genomes.

To exclude the probability of species misidentification, we collected all available GenBank *COI* sequences for the genera *Myoxocephalus* and *Megalocottus* (Appendix A) and, again, obtained two significantly diverged groups of sequences containing the *Myoxocephalus* and *Megalocottus* species (Figure 3; Cluster 1 and Cluster 2). It is important that the *My. quadricornis* sequences (MT303953 and MT303954) obtained in the present study are identical or very similar to the GenBank *COI* sequences (KJ128649, MG422966, EU524915, KX145213; Figure 3, see also Appendix A for additional sequences absent in Figure 3) reported by many other authors (Appendix A), which also rejects misidentification as the possible causative factor responsible for the observed close affinity between *My. quadricornis* and *Me. platycephalus*.

## 4. Discussion

The mt genome and nuclear rDNA data have revealed a mixed composition of the genus *Myoxocephalus*, presumably including also a *Megalocottus* species. The observed inconsistencies are supported by the GenBank *COI* sequences representing all the available *Myoxocephalus* and *Megalocottus* species. Both mitochondrial and nuclear markers indicate close evolutionary relationships between the genera *Myoxocephalus* and *Megalocottus* (as well as other studied sculpins belonging to the genera *Microcottus*, *Argyrocottus*, and *Porocottus*). Close associations between these genera were also inferred from morphological [5,6,13,14,15] and genetic [16,17,18,19,21,22,68] studies. However, the present genetic data does not support the synonymization of *Megalocottus* and *Myoxocephalus*, as was proposed by a number of authors (e.g., [10,11,69,70,71]). The genetic distance between them is significantly larger than the average interspecific distance within the genus *Myoxocephalus* (see above) and fits into the range of intergeneric distances detected for sculpin fishes [16,17,18], nevertheless, having the lowest values (see also the Results section). Instead, our genetic data indicate that the composition of the genera *Myoxocephalus* and *Megalocottus* needs revision. In particular, the closer affinity of *My. quadricornis* to the genus *Megalocottus* than to the genus *Myoxocephalus*, detected using both mt genomes and rDNA genes, strongly suggests the necessity of generic realignment by transferring *My. quadricornis* to the genus *Megalocottus*. The morphological, ecological, and life-history traits of the species do not contradict the genetic data obtained and also support the necessity of generic realignment. *My. quadricornis* and *Me. platycephalus* share a number of common features, including some ecological preferences and morphological traits, that distinguish them from the *Myoxocephalus* species [5,6,8,11,12]. In particular, the topography of the seismosensory system, which is a highly specific character used in fish taxonomy [6,12], is similar between *My. quadricornis* and *Me. platycephalus* on the one hand and differs them from the rest of the *Myoxocephalus* species on the other hand [6].

Our genetic data are consistent with the earlier reports of Cowan [12,72] who investigated relationships within the genus *Myoxocephalus* based on 45 morphological and 35 biochemical (isozymes) characters. The study placed *My. quadricornis* in the most basal position of the trees containing 13 *Myoxocephalus* species and constructed using morphological [12] and combined morphological and biochemical [72] data. Our results also agree with the data obtained by Knope [19], who showed that *Myoxocephalus* is not a monophyletic genus. The six *Myoxocephalus* species included in that study formed a monophyletic clade with high statistical support, except for one specimen identified as *Triglops quadricornis* (an unaccepted name of *My. quadricornis*; see, e.g., [4]), which grouped with a sister clade including *My. cf. scorpioides* and *Me. platycephalus* [19]. Smith and Busby [21] (p. 345), referring to a previous study by Kontula and Väinölä [25], indicated that “…*Myoxocephalus* may need minor revision.”

Using the *cytb*, *atp6*, and *atp8* mt genes, Kontula and Väinölä [25] found that the five marine *Myoxocephalus* species, included in their phylogenetic assessment, formed two main clades, while *My. quadricornis* was deeply nested within the *Myoxocephalus* genus. Based on these observations, the authors claimed that *My. quadricornis* (as well as its subspecies, the deepwater sculpin *My. q. thompsonii*) should be retained in *Myoxocephalus* without placing it in a separate genus, as it was proposed, for instance, by Neyelov [6]. Mecklenburg et al. [9] (Figure 4E, p. 118) and Mecklenburg and Steinke [28] (Figure 12, p. 176) obtained very similar tree topologies using the *COI* gene and also concluded that *My. quadricornis* should remain in *Myoxocephalus*.

The datasets of Kontula and Väinölä [25], Mecklenburg et al. [9], and Mecklenburg and Steinke [28] were limited only to the *Myoxocephalus* species and contained neither members of the genus *Megalocottus* nor other close sculpin genera (e.g., *Microcottus, Argyrocottus,* and *Porocottus*). The two divergent *Myoxocephalus* clades, revealed in their works, completely correspond to our Clusters 1 and 2 (Figure 1, Figure 2 and Figure 3). However, due to insufficient sample of species, the authors could not match the clades to representatives of *Myoxocephalus* and *Megalocottus* and recognize *Myoxocephalus* as a paraphyletic group containing both *Myoxocephalus* and *Megalocottus* species. When a dataset is represented by an insufficient sample of species, the hypothesis of monophyly can be mistakenly supported (see, e.g., [73], p. 298). Consequently, the conclusion of Kontula and Väinölä [25], Mecklenburg et al. [9], and Mecklenburg and Steinke [28] is not supported by our extended dataset containing species of fourteen genera (including *Myoxocephalus* and *Megalocottus*) and obtained based on both complete mt genomes (16.68 kb) and a long fragment of rDNA (6.33 kb), which represent more reliable markers than the short mtDNA fragments (0.65–1.98 kb) used by those authors.

Recently, Mecklenburg et al. [74] (Vol. 1, p. 210) noted that “DNA barcodes place *M. platycephalus* as a sister species to *Myoxocephalus quadricornis* in a larger all-inclusive clade of *Myoxocephalus* species”. The data obtained in the present study agrees with the first part of the statement made by Mecklenburg et al. [74]. Indeed, we show that *Me. platycephalus* and *My. quadricornis* are sister species, but we also show that *Myoxocephalus* is not an “all-inclusive clade”. Instead, it consists of two evolutionary distinct lineages at the intergeneric level of divergence (see Figure 1, Figure 2 and Figure 3), which is clearly evidenced by a comparative analysis of interspecific and intergeneric distances for *Myoxocephalus*, *Megalocottus*, and other sculpins (see the Results section). No such analyses were reported in the works of Kontula and Väinölä [25], Mecklenburg et al. [9,74], and Mecklenburg and Steinke [28].

The observed discordance may be explained in part by incorrect taxonomic identification, as was noted by McCusker et al. [29]. For instance, Smith and Busby [21] reported that *Myoxocephalus* was paraphyletic relative to *Microcottus*. We reanalyzed Smith and Busby’s data and found that the *Mi. sellaris* specimen was misidentified and actually represented *My. scorpius*. The very low distance (0.86%) between the *My. scorpius CytB* nucleotide sequence (GenBank accession number MK321578) and the corresponding sequence (KM057906) of Smith and Busby’s “*Microcottus*” clearly indicates that these authors misidentified *Mi. sellaris*. The two other sequences of *Mi. sellaris* obtained independently by Togashi [75] (LC125750) and Balakirev, Kravchenko, and Semenchenko (present study) show a very substantial divergence (around 11%) between *Mi. sellaris* and *My. scorpius,* which explains the clear resolution of these species (Figure 1). Thus, the observation of Smith and Busby [21] that *Microcottus* is nested within *Myoxocephalus* is erroneous, being based on incorrect species identification (other examples of incorrect taxonomic identification see, e.g., in [28,33]). Another problem potentially causing different tree topologies arises from DNA barcoding, which is still used and recommended for fish identifications [9,27,28,29]. However, the DNA barcoding approach is frequently not enough to obtain reliable resolution. This problem is highly debated (for recent reviews see, e.g., [76,77]).

In accordance with the principle of priority in the International Code of Zoological Nomenclature (ICZN; Article 23) [78], Neyelov [6] proposed to place *My. quadricornis* in a separate monotypic genus, *Triglopsis* Girard 1851 (which is currently an unaccepted junior synonym of the genus *Myoxocephalus* Tilesius 1811 [2,3,4]), based mostly on the seismosensory system topography and other morphological characters. In this case, *Me. platycephalus* should also be transferred to the genus *Triglopsis* due to the genetic data obtained in the present work and the principle of priority, because *Triglopsis* Girard 1851 precedes *Megalocottus* Gill 1861. However, this hypothetic transfer contradicts the classification of Neyelov [6], who, in spite of the close relationship between *My. quadricornis* and *Me. platycephalus* [5,6,8,11,12], did not include both species in the same genus *Triglopsis* [6].

The type species of the genus *Triglopsis* is the deepwater sculpin *T. thompsonii* Girard 1851 (currently a subspecies of *My. quadricornis* [25]) inhabiting the North American continental lakes. This sculpin is a highly specialized form adapted to freshwater habitats. It originated from an Arctic marine lineage of the fourhorn sculpin *My. quadricornis* that was driven south to freshwater habitats by early glacial advances within the Early to Middle Pleistocene or earlier [79,80]. The divergence time between *T. q. thompsonii* and *My. quadricornis* is estimated at around one million years (Myr) ago [25,81] based on the *COI*, *cytb*, *atp6*, and *atp8* mt genes. In turn, the Arctic *My. quadricornis* lineage diverged from the ancient *Myoxocephalus* forms from the Pacific, including closely related *Megalocottus* lineage, in the late Pliocene (see [5,6,14,82] and references therein), which agrees well with the estimated divergence time (≈7.9 Myr ago) between the Pacific and Arctic–Atlantic *Myoxocephalus* species [81].

In accordance with the principle of priority in the International Code of Zoological Nomenclature (ICZN; Article 23) [78], both species, *My. quadricornis* and *Me. platycephalus*, should be placed in the genus *Triglopsis* Girard 1851. However, this would contradict the basic assumption of phylogenetic taxonomy [73,83,84], because *Triglopsis* represents a derived form relative to the ancestral *Myoxocephalus* and *Megalocottus* forms. Thus, at the moment, we find it unreasonable to resurrect the genus *Triglopsis* for both *My. quadricornis* and *Me. Platycephalus*; instead, we consider the transfer of *My. quadricornis* to the genus *Megalocottus* as a feasible alternative, which is consistent with the evolutionary history of both Pacific (by origin) forms [5,6,14,82].

Taking in account the above circumstances, it would be premature to provide the diagnosis, description, and classification of *My. quadricornis* within the traditional Linnaean system. The compositions of the genera *Myoxocephalus* and *Megalocottus* inferred in the present work are delimited based on a phylogenetic approach according to the concept of phylogenetic systematics [73,84]. Further studies using complete mt genomes and nuclear genes, along with a more detailed morphological analysis and representative sample of species, are necessary to build a natural classification of these sculpin fishes that would satisfy the requirements of both the traditional Linnaean system and the phylogenetic systematics.

## 5. Conclusions

The present data, obtained through the analysis of pairwise genetic distances and phylogenetic reconstructions with high bootstrap support, indicates an intergeneric level of divergence between *Megalocottus*, *Myoxocephalus*, *Microcottus*, *Argyrocottus*, and *Porocottus*. It also shows some obvious inconsistencies in the relationships among the sculpin fishes belonging to the genera *Myoxocephalus* and *Megalocottus* (Figure 1, Figure 2 and Figure 3), which has always been a controversial and long-debated issue [6,9,10,11,28,71,74]. The multi-gene approach provides evidence for a close affinity of *My. quadricornis* with the species of the genus *Megalocottus* (but not with *Myoxocephalus*) and strongly suggests *My. quadricornis* to be placed in the genus *Megalocottus*.

We believe that the proposed taxonomic realignment based upon both mitochondrial and nuclear genes is robust. However, we also expect that the genera *Myoxocephalus* and *Megalocottus*, as well as other cottids, will continue to be subject to taxonomic changes in the future. In particular, the GenBank *COI* sequences suggest that three other *Myoxocephalus* species (besides *My. quadricornis*), *My. aenaeus*, *My. octodecemspinosus*, and *My. scorpioides*, may also belong to the genus *Megalocottus* (and, consequently, be removed from the genus *Myoxocephalus*; see above). To resolve this issue, a large-scale genetic analysis including complete mt genomes and nuclear genes is necessary. The next-generation sequencing (NGS) provides extensive full nuclear genome data, which is highly efficient for identifying speciation events and taxonomic relationships among non-model fish species. It turned out that a few candidate adaptive genes may demonstrate high divergence and create a barrier for the gene flow leading to reproductive isolation between species, which may be identical for the rest part of the nuclear genome [85,86,87,88]. A targeted NGS approach makes it possible to identify a suite of loci that can be used in future research to investigate the genomic basis of adaptation, differentiation, speciation, and phylogenetic relationships between sculpins, the most challenging group of fish for genetic and taxonomic studies.

## Figures and Tables

**Figure 1 genes-11-01071-f001:**
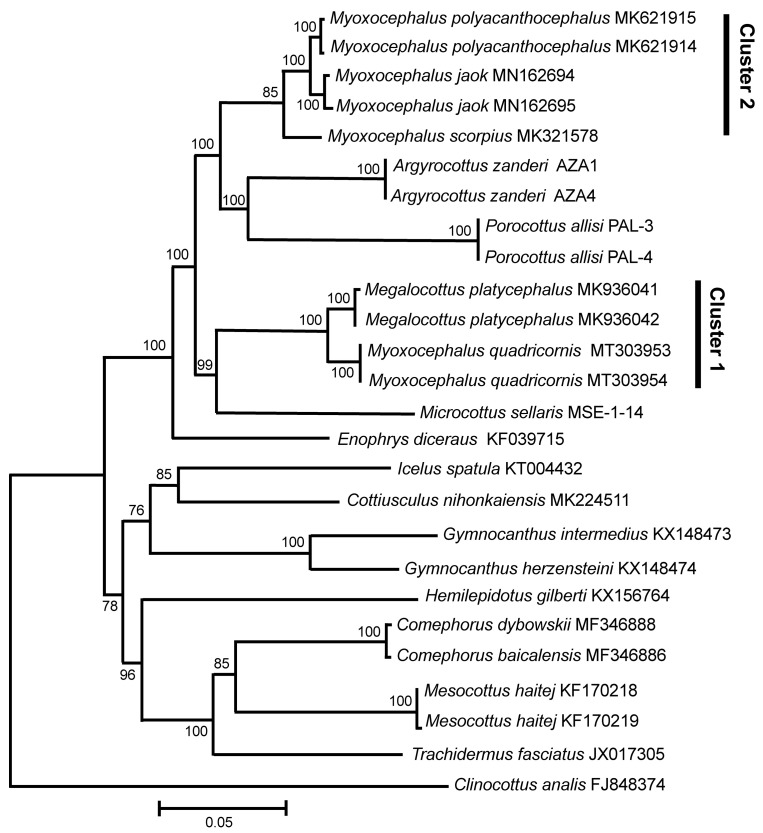
Maximum likelihood tree for the fourhorn sculpin *Myoxocephalus quadricornis* and members of the family Cottidae from GenBank inferred from the complete mitochondrial genomes. The tree is based on the General Time Reversible + gamma + invariant sites (GTR + G + I) model of nucleotide substitution. The numerals at the nodes are bootstrap percentage probability values based on 1000 replications (values below 75% are omitted).

**Figure 2 genes-11-01071-f002:**
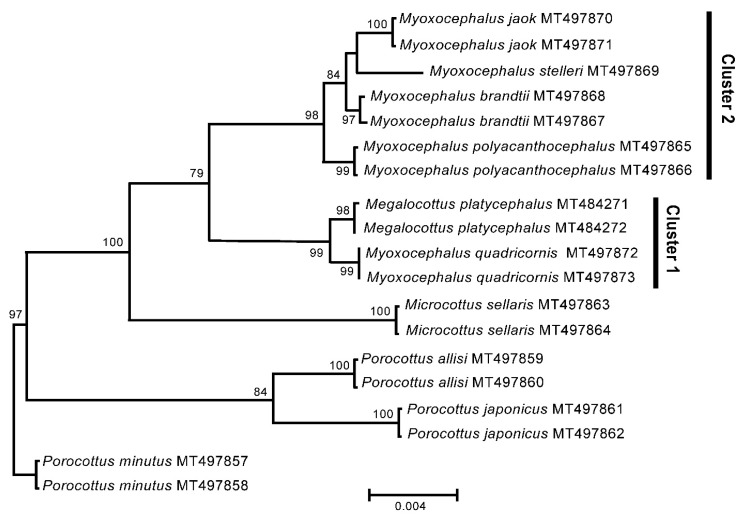
Maximum likelihood tree for the fourhorn sculpin *Myoxocephalus quadricornis* and members of the family Cottidae from GenBank inferred from the 6332 bp ribosomal DNA. The tree is based on the Tamura-Nei (TN93) model of nucleotide substitution. For other comments, see Figure 1.

**Figure 3 genes-11-01071-f003:**
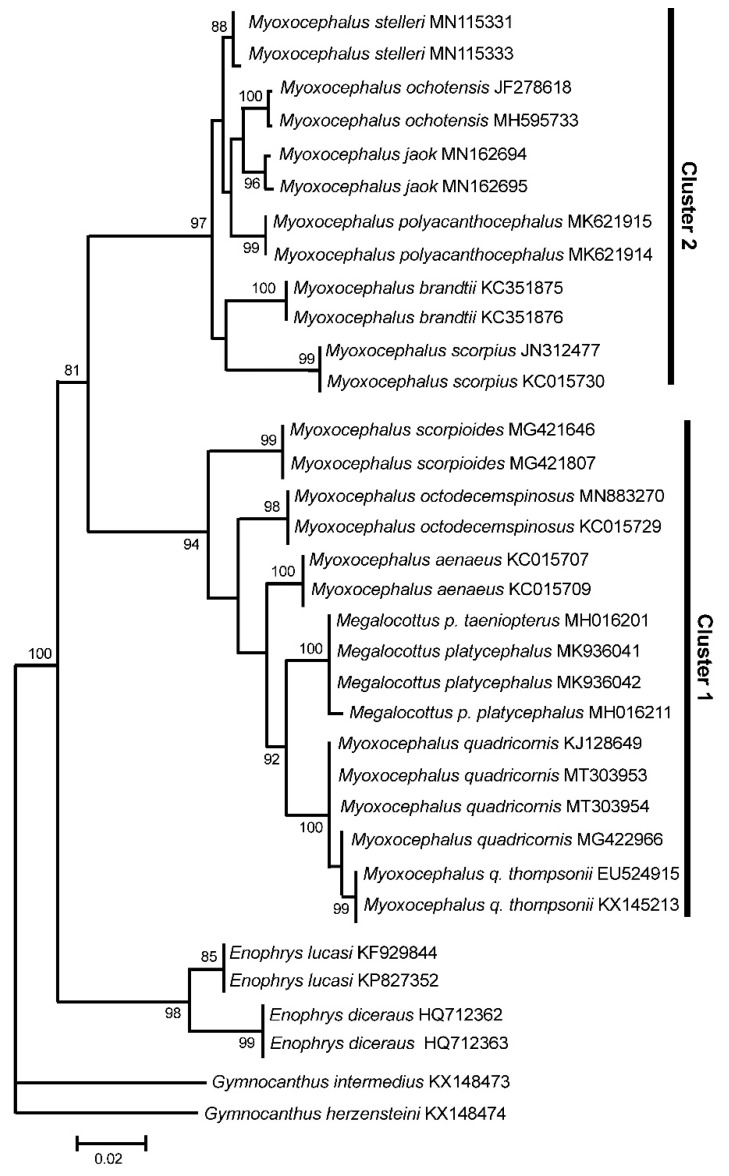
Maximum likelihood tree for the fourhorn sculpin *Myoxocephalus quadricornis* and members of the family Cottidae from GenBank inferred from the mitochondrial *COI* gene. The tree is based on the Hasegawa-Kishino-Yano + gamma + invariant sites (HKY + G + I) model of nucleotide substitution. Species of the genera *Enophrys* and *Gymnocanthus* are used as outgroups. For other comments, see Figure 1.

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
