# Peer review of "Genetic Evidence for a Mixed Composition of the Genus Myoxocephalus (Cottoidei: Cottidae) Necessitates Generic Realignment"

_genes, 2020, doi:10.3390/genes11091071_

Round 1
Reviewer 1 Report
The manuscript entitled “Genetic evidence for a mixed composition of the genus Myoxocephalus (Cottoidei: Cottidae) necessitates generic realignment” by Balakirev and coauthors reports the phylogenetic relationships of Myxocephalus and Megalocottus species. The study is the continuation of previous studies of this and other research groups on sculpin fishes, a widely distributed group of euryhaline fishes and a promising targets for evolutionary analyses because of their fast speciation rates among fishes. This study attempts to resolve major cottoid phylogenies from multiple molecular datasets, i.e. nuclear ribosomal DNA, a traditional barcoding marker like COI, and complete mitogenomes, the latter being the main methodological advancement for cottoid phylogeny. All molecular trees agree on the close relationships of some Myoxocephalus species, in primis Myoxocephalus quadricornis, with the species of the genus Megalocottus. The long standing issue of phylogenetic conflict and taxonomic uncertainty among Cottoidea fish is likely due to the evolutionary radiation of a very large teleost suborder (more than 800 species). Although I don't consider myself sufficiently informed on cottoid fish phylogeny to comment on some of the theoretical aspects of this contribution, it is clear that the major aspects in this manuscript concern the comparative context -the set up, the assembly, of the argument concerning the evolutionary questions in the first place, and then the claimed significance of these data in the results and discussion. The main conclusions of this manuscript are that “…the composition of the genera Myoxocephalus and Megalocottus needs revision” (lines 306-307) and “..the transfer of My. quadricornis to the genus Megalocottus…” (lines 383-384). Although this is not a novelty in itself, since other authors reached similar conclusions in pre-molecular times, this is the first demonstration based on multiple molecular datasets including mitogenomes. Finally, the manuscript is clear and well written, with no fundamental flaws and weaknesses, andcontains new and interesting data that are sound, adequately described and illustrated, and that may provide important cues to scientists interested in the Cottoidea fish evolution field. Therefore, after amending the manuscript according to the following suggestions, it is suitable for publication in Genes.
General comment:
L103-104: Concerning one of the two paper objectives, i.e. why it is important to study Megalocottus-Myoxocephalus intergeneric relationships, the argument of the close affinity of these two genera could be more comprehensively introduced by citing not only morphoanatomical studies (L61-63), but also the Knope’s 2013 molecular paper (lines 383-384 of Results section).
Minor revisions:
L94: it would be useful to know more on why sculpin barcoding is challenging based on McCusker’s work.
Figure 3: in the tree, Megalocottus platycephalus should be written as subspecies (Megalocottus p. platycephalus) as in case of Megalocottus p. taeniopterus.
L.114: Me. platycephalus
L273 and 325: why cf. in My. cf. scorpioides?
L347: Me. platycephalus
Table S1: Myoxocephalus brandtii
Text S1: COI in italics
Author Response
Responses to the Reviewers
Reviewer 1:
General comment:
L103-104: Concerning one of the two paper objectives, i.e. why it is important to study Megalocottus-Myoxocephalus intergeneric relationships, the argument of the close affinity of these two genera could be more comprehensively introduced by citing not only morphoanatomical studies (L61-63), but also the Knope’s 2013 molecular paper (lines 383-384 of Results section).
Response to Reviewer 1: Done. Thanks.
We have added the Knope’s 2013 molecular paper to Introduction (lines 64-67, revision).
Reviewer 1:
Minor revisions:
L94: it would be useful to know more on why sculpin barcoding is challenging based on McCusker’s work.
Response to Reviewer 1: Done. Thanks.
We have extended the respective sentence (lines 94-97, revision) as follows: McCusker et al. [23] noted, however, that sculpins (along with eelpouts and rocklings) proved to be among the most challenging groups for DNA barcoding, which could be explained by a number of factors including incomplete lineage sorting, introgressive hybridization, or erroneous taxonomic identifications.
Reviewer 1: Figure 3: in the tree, Megalocottus platycephalus should be written as subspecies (Megalocottus p. platycephalus) as in case of Megalocottus p. taeniopterus.
Response to Reviewer 1: The species and subspecies names that we use are in a strict accordance with the names used by the corresponding authors who provided the nucleotide sequences.
Reviewer 1: L.114: Me. platycephalus
Response to Reviewer 1: Done. Thanks. (L117, revision)
Reviewer 1: L273 and 325: why cf. in My. cf. scorpioides?
Response to Reviewer 1: In biological naming conventions, cf. is commonly placed between the genus name and the species name to denote a specimen that is hard to identify because of practical difficulties, such as poor fixation.
Reviewer 1: L347: Me. platycephalus
Response to Reviewer 1: Done. Thanks (lines 360-361, revision).
Reviewer 1: Table S1: Myoxocephalus brandtii
Response to Reviewer 1: Done. Thanks.
Reviewer 1: Text S1: COI in italics
Response to Reviewer 1: Done. Thanks.
Reviewer 2 Report
Overall, this is an interesting study that is well written. The analyses and data are sound (with one possible concern about the analyses in Figure 3). My concerns with the paper are in some of their attention to detail and over reliance on pairwise distances.
I have four fundamental issues with the paper that need to be corrected throughout, but I will highlight these areas that need revision.
1) Throughout the paper the authors project a knowledge base where the systematics of cottoids is in complete disarray, and I actually think that the relationships among the sculpins are comparatively well resolved by the standards of many perciform groups. This issue is perhaps best highlighted in the abstract. Contrary to the statements in the manuscript, the work of the authors of this paper, Buser, Goto, Jackson, Kinziger, Knope, Kontula, Smith, and Yabe (and their collaborators) have really improved our understanding of the relationships among cottoids to the point that it is simply FALSE that the relationships of sculpins remain largely unresolved. The amount of congruence and complementarity far outweighs the problems.
2) This problem is best highlighted in the first 120 lines of the manuscript, but the manuscript's treatment of Microcottus and the close relatives of Myoxocephalus is at best imprecise and at worst misleading. In lines 60–62, there is plenty of evidence that suggests that Myoxocephalus is not most closely related to Megalocottus or that it is at least a lot more complicated. Yabe's work suggests that Microcottus, Porocottus, and Argyrocottus are closely related. Knope's (MP&E) work suggests that Triglops [probably incorrect genus assignment], Microcottus, and Megalocottus are most closely related and nested within the genus. Smith and Busby's (MP&E) work suggests that Microcottus is nested within Myoxocephalus. All of these studies and their results need to be cited and discussed beyond the citations noted by the authors, which are not the most appropriate citations at all for these statements.
3) The ambiguity in point 2 is made more complicated in the paper because the authors sometimes treat Microcottus sellaris as a member of Microcottus, but sometimes treat it as a member of Myoxocephalus. I am unaware of anyone treating Microcottus sellaris as Myoxocephalus (not that I would disagree with that assignment, but it would need to be formally done), but these authors change its generic treatment between lines 101 and 114 and between line 114 and Figure 2 without discussion. This seems particularly strange given that a major result of the paper is about recognizing Triglopsis (as a separate genus). More to the point, why wasn't Microcottus sellaris included in the the COI analysis given that sequences for this species are in GenBank? The placement and generic placement of Microcottus is absolutely something that should be resolved in this study. Including the available COI sequences in the analysis presented in Figure 3 seems mandatory.
4) Finally, and perhaps most importantly, I do not support or understand the authors emphasis on distance for arguing for the treatment of cluster 1 is more than one genus. It seems truly clear that cluster two should retain the name Myoxocephalus. The question should be whether cluster 1 and 2 should both be Myoxocephalus (possibly/probably including Microcottus sellaris within the genus) or whether cluster 1 and cluster 2 should be independent genera (placement of Microcottus would need to be resolved by adding it to figure 3). All that being said, it is hard to imagine justifying the recognition of a monotypic genus Triglopsis sister to a monotypic genus Megalocottus as suggested by the authors. As Hennig first argued, a major goal of phylogenetics is to produce classifications that reflect phylogenetic relationships. Two monotypic genera sister to each other provides no evidence of common ancestry relative to other sculpin genera.
Other than those four points, I think the manuscript is in good shape. These four concerns are very real, but manageable changes that are necessary.
Author Response
Responses to the Reviewers
Reviewer 2: “I have four fundamental issues with the paper that need to be corrected throughout, but I will highlight these areas that need revision.
1) Throughout the paper the authors project a knowledge base where the systematics of cottoids is in complete disarray, and I actually think that the relationships among the sculpins are comparatively well resolved by the standards of many perciform groups. This issue is perhaps best highlighted in the abstract. Contrary to the statements in the manuscript, the work of the authors of this paper, Buser, Goto, Jackson, Kinziger, Knope, Kontula, Smith, and Yabe (and their collaborators) have really improved our understanding of the relationships among cottoids to the point that it is simply FALSE that the relationships of sculpins remain largely unresolved. The amount of congruence and complementarity far outweighs the problems.”
Response to Reviewer 2: Done. Thanks.
We have deleted the sentence (line 30, revision) “The evolutionary relationships and taxonomy of sculpins still remain largely unresolved.” From Abstract.
Reviewer 2: “2) This problem is best highlighted in the first 120 lines of the manuscript, but the manuscript's treatment of Microcottus and the close relatives of Myoxocephalus is at best imprecise and at worst misleading. In lines 60–62, there is plenty of evidence that suggests that Myoxocephalus is not most closely related to Megalocottus or that it is at least a lot more complicated. Yabe's work suggests that Microcottus, Porocottus, and Argyrocottus are closely related. Knope's (MP&E) work suggests that Triglops [probably incorrect genus assignment], Microcottus, and Megalocottus are most closely related and nested within the genus. Smith and Busby's (MP&E) work suggests that Microcottus is nested within Myoxocephalus. All of these studies and their results need to be cited and discussed beyond the citations noted by the authors, which are not the most appropriate citations at all for these statements.”
Response to Reviewer 2: Done. Thanks.
We have additionally assembled and analyzed four complete mitochondrial genomes for Argyrocottus zanderi and Porocottus coronatus, and the most (12,442 bp) of the Microcottus sellaris mitochondrial genome. These new data are incorporated in Figure 1.
The obtained results are consistent with the Yabe's work (1985) and suggest Microcottus, Porocottus, and Argyrocottus to be closely related (Figure 1).
Smith and Busby (2014) reported that Myoxocephalus is paraphyletic relative to Microcottus (“Smith and Busby's (MP&E) work suggests that Microcottus is nested within Myoxocephalus”). We have reanalyzed the Smith and Busby's data and found that the Microcottus sellaris specimen was misidentified and actually represented Myoxocephalus scorpius (see the tree below).
The values of genetic distances are in full accordance with the tree clustering (see below).
[1] #Smith_Busby_2014
[2] #Myoxocephalus scorpius
[3] #Microcottus sellaris Togashi_2019
[4] #Microcottus sellaris Balakirev_Kravchenko_Semenchenko
[ 1 2 3 4 ]
[1] [ 0.0038 ][ 0.0126 ][ 0.0129 ]
[2] 0.0086 [ 0.0128 ][ 0.0131 ]
[3] 0.1082 0.1100 [ 0.0029 ]
[4] 0.1100 0.1117 0.0052
The very low distance (0.86%) between Myoxocephalus scorpius and Smith and Busby's “Microcottus” clearly indicates that these authors misidentified Microcottus sellaris. The two other sequences of Microcottus sellaris obtained independently by Togashi (2019, unpublished) and Balakirev, Kravchenko, and Semenchenko (present data) show a very substantial divergence (around 11%) from Myoxocephalus scorpius.
Thus, the suggestion of Smith and Busby (MP&E) that Microcottus is nested within Myoxocephalus is not correct, being based on erroneous species identification. Unfortunately, we could not verify the data of Knope (2013) concerning the same issue, because no respective nucleotide sequences of Microcottus are available in GenBank. This, however, not critically important because our dataset is confirmed by an independent resource (Togashi’s sequences).
The papers of Knope (2013) and Smith and Busby (2014) are cited and discussed (lines 331-337, 379-395, revision).
In accordance with the Reviewer’s 2 comment, we have changed the sentence (lines 64-65) from “Among all members of the family Cottidae, the genus Myoxocephalus is most closely related to the genus Megalocottus Gill 1861…” to “Based on the morpho-anatomical [5,6,14-15] and genetic [55-57,61] studies it was shown that the genus Myoxocephalus is closely related to the genus Megalocottus Gill 1861 …”
Reviewer 2: “3) The ambiguity in point 2 is made more complicated in the paper because the authors sometimes treat Microcottus sellaris as a member of Microcottus, but sometimes treat it as a member of Myoxocephalus. I am unaware of anyone treating Microcottus sellaris as Myoxocephalus (not that I would disagree with that assignment, but it would need to be formally done), but these authors change its generic treatment between lines 101 and 114 and between line 114 and Figure 2 without discussion. This seems particularly strange given that a major result of the paper is about recognizing Triglopsis (as a separate genus). More to the point, why wasn't Microcottus sellaris included in the the COI analysis given that sequences for this species are in GenBank? The placement and generic placement of Microcottus is absolutely something that should be resolved in this study. Including the available COI sequences in the analysis presented in Figure 3 seems mandatory.”
Response to Reviewer 2: We do not “…treat Microcottus sellaris as a member of Microcottus, but sometimes treat it as a member of Myoxocephalus.” The last paragraph of the Introduction section (lines 98-108, revision) contains just a list of species studied in the present work without their phylogenetic characteristics.
The mtDNA and nuclear genetic data show that Microcottus (as well as Argyrocottus and Porocottus) do not belong to Myoxocephalus (Figure 1). This conclusion is supported by the analysis of genetic distances and phylogenetic reconstructions with high bootstrap support (Figures 1 and 2).
We have not changed Figure 3 (COI data), because the relationships between Microcottus, Porocottus, Argyrocottus, Myoxocephalus, and Megalocottus are resolved more reliably with the much longer mitochondrial genomes (Figure 1), which we have additionally obtained for this revision.
Reviewer 2: “4) Finally, and perhaps most importantly, I do not support or understand the authors emphasis on distance for arguing for the treatment of cluster 1 is more than one genus. It seems truly clear that cluster two should retain the name Myoxocephalus. The question should be whether cluster 1 and 2 should both be Myoxocephalus (possibly/probably including Microcottus sellaris within the genus) or whether cluster 1 and cluster 2 should be independent genera (placement of Microcottus would need to be resolved by adding it to figure 3). All that being said, it is hard to imagine justifying the recognition of a monotypic genus Triglopsis sister to a monotypic genus Megalocottus as suggested by the authors. As Hennig first argued, a major goal of phylogenetics is to produce classifications that reflect phylogenetic relationships. Two monotypic genera sister to each other provides no evidence of common ancestry relative to other sculpin genera.”
Response to Reviewer 2: Done. Thanks.
We have deleted Table 1 containing the values of genetic distances.
Genetic distance is a measure of genetic divergence between species or populations. It is a useful metric allowing a quantitative comparison between the taxa under study (e.g., Nei 1987a,b; Avise 2004).
We focus on distance for arguing that the treatment of cluster 1 is lower than two genera because the level of divergence between My. quadricornis and Me. platycephalus matches the average value of divergence between different species within the same genus (intragenus level).
Our data on mt genomes and rDNA sequences suggest that clusters 1 and 2 should be independent genera (Figure 1). The data also indicate that Myoxocephalus does not include Microcottus sellaris (Figures 1 and 2). If we accept the first hypothesis (“…whether cluster 1 and 2 should both be Myoxocephalus (possibly/probably including Microcottus sellaris within the genus…”), then we should extend Myoxocephalus by including at least three other genera (Argyrocottus, Porocottus, and Enophrys) because genetic distances between them are of the same order of magnitude (lines 220-223, revision; Figure 1). Instead of this scenario, we suggest that Myoxocephalus, Megalocottus, Microcottus, Argyrocottus, Porocottus, and Enophrys, should be recognized as independent genera. Further study based on the targeted NGS approach, providing multiple sensitive markers, is necessary to clarify phylogenetic relationships between them.
Phylogenetic relationships between the sculpins mentioned above are, however, not a main focus of our study. As we formulate in the Introduction section (lines 105-108), there are “…two main objectives: (1) investigate the phylogenetic affinities of different species of Myoxocephalus and (2) clarify the Myoxocephalus–Megalocottus intergeneric relationships within the family Cottidae, which would allow assessment of circumscription for both genera.” These objectives were motivated by the conflicting relationships between Myoxocephalus quadricornis and the rest of the species belonging to this genus. Phylogenetic analyses showed that Myoxocephalus quadricornis is clustered with Megalocottus platycephalus but not with the Myoxocephalus species. Consequently, Myoxocephalus quadricornis is closer to clade Megalocottus than to clade Myoxocephalus.
References
Avise J.C. (2004). Molecular Markers, Natural History, and Evolution. Sinauer Associates: Sunderland, MA.
Nei M. (1987a). Molecular Evolutionary Genetics. New York: Columbia University Press.
Nei M. (1987b) Genetic distance and molecular phylogeny. In: Population Genetics and Fishery Management (N. Ryman and F. Utter, eds.), University of Washington Press, Seattle, WA, pp. 193–223.
